# Prescriptive analytics for reducing 30-day hospital readmissions after general surgery

**Dimitris Bertsimas**[1], **Michael Lingzhi Li**[1], **Ioannis Ch. Paschalidis**[2]*, **Taiyao Wang**[2]

**1** Operations Research Center, Massachusetts Institute of Technology, Cambridge, MA, United States of America, **2** Center for Information and Systems Engineering, Boston University, Boston, MA, United States of America

* yannisp@bu.edu

## Abstract

**Data Availability Statement:** Regarding data availability, we unfortunately cannot share the data used in our study because we are bound by data use agreements. The data are available to participant institutions through the program

### Introduction

New financial incentives, such as reduced Medicare reimbursements, have led hospitals to closely monitor their readmission rates and initiate efforts aimed at reducing them. In this context, many surgical departments participate in the American College of Surgeons National Surgical Quality Improvement Program (NSQIP), which collects detailed demographic, laboratory, clinical, procedure and perioperative occurrence data. The availability of such data enables the development of data science methods which predict readmissions and, as done in this paper, offer specific recommendations aimed at preventing readmissions.

### Materials and methods

This study leverages NSQIP data for 722,101 surgeries to develop predictive and prescriptive models, predicting readmissions and offering real-time, personalized treatment recommendations for surgical patients during their hospital stay, aimed at reducing the risk of a 30-day readmission. We applied a variety of classification methods to predict 30-day readmissions and developed two prescriptive methods to recommend pre-operative blood transfusions to increase the patient's hematocrit with the objective of preventing readmissions. The effect of these interventions was evaluated using several predictive models.

### Results

Predictions of 30-day readmissions based on the entire collection of NSQIP variables achieve an out-of-sample accuracy of 87% (Area Under the Curve—AUC). Predictions based only on pre-operative variables have an accuracy of 74% AUC, out-of-sample. Personalized interventions, in the form of pre-operative blood transfusions identified by the prescriptive methods, reduce readmissions by 12%, on average, for patients considered as candidates for pre-operative transfusion (pre-operative hematoctic <30). The prediction accuracy of the proposed models exceeds results in the literature.

described at https://www.facs.org/quality-programs/acs-nsqip. So, in principle, any hospital which participates in the ACS NSQIP program could obtain access to the data. The authors did not have any special access privileges to the data other NSQIP participants would not have.

**Funding:** ICP was supported by the US National Science Foundation (https://www.nsf.gov/) under grants IIS-1914792, DMS-1664644, and CNS-1645681, by the Office of Naval Research (https://www.onr.navy.mil/) under MURI grant N00014-19-1-2571, by the National Institutes of Health (https://www.nih.gov/) under grant R01 GM135930 and grant UL54 TR004130 to the Clinical & Translational Science Institute at Boston University, and by the Boston University Digital Health Initiative (https://www.bu.edu/ihsip/). The funders had no role in study design, data collection and analysis, decision to publish, or preparation of the manuscript.

**Competing interests:** The authors have declared that no competing interests exist.

## Conclusions

This study is among the first to develop a methodology for making specific, data-driven, personalized treatment recommendations to reduce the 30-day readmission rate. The reported predicted reduction in readmissions can lead to more than $20 million in savings in the U.S. annually.

## 1. Introduction

The United States spends $3 trillion annually on healthcare, corresponding to more than 17% of the U.S. GDP and far exceeding the next-highest spender among high-income countries [1]. While many factors contribute to higher spending, hospital readmissions, defined as an additional admission to address the same issue within 30 days after discharge, are an important – and potentially preventable –source of excessive resource utilization [2, 3].

In an effort to reduce unnecessary costs, the Affordable Care Act of 2012 introduced financial penalties for hospitals with readmission rates above the national average. While these measures have so far concentrated on medical conditions (e.g., acute myocardial infarction, congestive heart failure, pneumonia) and common orthopedic procedures (e.g., hip and knee arthroplasty), the list could expand to include common general surgical procedures.

In anticipation of these changes, Surgical Departments have started to closely monitor their readmission rates, and establishing processes aimed at reducing them. Several authors have sought to determine common causes of readmission after general surgical procedures, and most appear to relate to pre-existing conditions [4–7] and complications after surgery [8].

In 2005, the American College of Surgeons (ACS) established the National Surgical Quality Improvement Program (NSQIP), which collects detailed demographic, laboratory, clinical, procedure and perioperative occurrence data, currently for General Surgery, and eventually in several subspecialties. The availability of such data, enables the development of *data analytics* methods relevant to the readmission reduction efforts.

While earlier work has primarily focused on readmission *predictive methods*, there has only been limited attention given to specific interventions with the potential to reduce readmissions; and that has focused mostly on post-discharge care [9–11]. Earlier work on predictive methods for hospitalizations have been successful but focused on specific diseases [12–14].

The objective of this work is to develop more direct *prescriptive methods* that offer specific treatment recommendations during the patients' hospital stay with the potential to reduce readmission risk. Our recommended interventions are driven by data; essentially, for each patient, we learn from data what has been effective in preventing a readmission for other "similar" patients. While the methodologies we develop are general and can be applicable to any sort of interventions, we focus on in-hospital treatment because the NSQIP data we leverage contain only such variables. We further focus on the patients' pre-operative hematocrit because it is commonly measured, important for assessing readmission risk, and easily modulated through blood transfusion.

## 2. Materials and methods

### 2.1 Data and pre-processing

The ACS-NSQIP dataset we use in our analysis contains over 300 variables on comorbidities, intra-operative events, and 30-day outcomes using prospective random sampling [15],

**Table 1. Assumed baseline treatment.**

| Assumed Transfusion Factuals | Condition in Data |
|---|---|
| No blood transfusion | TRANSFUS = 0 |
| 1 bag of blood | HCT>30 and TRANSFUS = 1 |
| 2 bags of blood | 27<HCT<30 and TRANSFUS = 1 |
| 3 bags of blood | HCT<27 and TRANSFUS = 1 |

including: (i) baseline demographic and health care status characteristics (e.g., age, gender, race, BMI, smoking, diabetes, hypertension, admittance from the ER); (ii) procedure information (e.g., CPT codes, ICD9 codes, ASA classification, wound classification); (iii) pre-operative, intra-operative, and post-operative variables, such as hospital stay information, Surgical Site Infections (SSI, superficial/deep/organ space) and complications (e.g., pneumonia, infections, bleeding, thromboembolic events), and (iv) laboratories, including pre-operative and post-operative values.

The NSQIP dataset at our disposal included more than 2.2 million surgeries during 2011–2014. While the NSQIP program provided high-quality manually curated data obtained from trained data abstractors, the variable definitions change over time. Specifically, the definitions of the occurrences listed in Table 2 (e.g., sepsis, pneumonia, SSIs) have

**Table 2. Most statistically significant differences in readmitted and non-readmitted patients.**

| Variable | All patients | Readmitted | Non-Readmitted | p-value |
|---|---|---|---|---|
| Estimated Probability of Morbidity | 0.06 | 0.11 | 0.06 | <0.000001 |
| Pre-operative hematocrit | 39.67 | 37.85 | 39.78 | <0.000001 |
| The American Society of Anesthesiology (ASA) Physical Status Classification | 2.43 | 2.78 | 2.4 | <0.000001 |
| Estimated Probability of Mortality | 0.01 | 0.02 | 0.01 | <0.000001 |
| Total operation time in minutes | 111.31 | 148.79 | 109.14 | <0.000001 |
| Return to OR (binary) | 0.03 | 0.24 | 0.02 | <0.000001 |
| Number of Superficial Wound Occurrences | 0.02 | 0.08 | 0.01 | <0.000001 |
| Number of Deep Incisional SSI Occurrences | 0.01 | 0.06 | 0 | <0.000001 |
| Number of Organ/Space SSI Occurrences | 0.01 | 0.11 | 0.01 | <0.000001 |
| Number of Urinary Tract infection Occurrences | 0.01 | 0.06 | 0.01 | <0.000001 |
| Number of Bleeding Transfusions Occurrences | 0.06 | 0.13 | 0.05 | <0.000001 |
| Number of Sepsis Occurrences | 0.02 | 0.1 | 0.01 | <0.000001 |
| Days from Operation to Discharge | 2.77 | 4.51 | 2.67 | <0.000001 |
| OUTPATIENT (if surgical procedure was performed in an outpatient setting) | 0.4 | 0.18 | 0.42 | <0.000001 |
| CPT_Muscl_29x: Casts and endoscopy/arthroscopy | 0.03 | 0.01 | 0.03 | <0.000001 |
| Indicator for any morbidity/complications | 0.12 | 0.49 | 0.1 | <0.000001 |
| no diagnosis of diabetes or diabetes controlled by diet alone. | 0.85 | 0.77 | 0.85 | <0.000001 |
| Discharge Destination: Home | 0.9 | 0.82 | 0.91 | <0.000001 |
| Pre-operative alkaline phosphatase | 69.37 | 82.75 | 68.59 | <0.000001 |
| Pre-operative serum albumin | 3.95 | 3.78 | 3.96 | <0.000001 |
| ICD9 550: Inguinal hernia | 0.04 | 0.01 | 0.04 | <0.000001 |
| Work Relative Value Unit (a metric of surgical complexity) | 16.35 | 19.75 | 16.16 | <0.000001 |
| Age | 56.41 | 60.42 | 56.17 | <0.000001 |
| Hypertension requiring medication | 0.45 | 0.57 | 0.44 | <0.000001 |
| Elective Surgery (binary) | 0.8 | 0.69 | 0.81 | <0.000001 |

(*Continued*)

**Table 2.** (Continued)

| Variable | All patients | Readmitted | Non-Readmitted | p-value |
|---|---|---|---|---|
| Number of Pneumonia Occurrences | 0.01 | 0.05 | 0.01 | <0.000001 |
| Bleeding disorders | 0.04 | 0.09 | 0.04 | <0.000001 |
| Open wound/wound infection | 0.03 | 0.07 | 0.03 | <0.000001 |
| Number of DVT/Thrombophlebitis Occurrences | 0.01 | 0.04 | 0 | <0.000001 |
| CPT_Muscl_23x-25x: Shoulder arm wrist hand | 0.03 | 0.01 | 0.03 | <0.000001 |
| CPT_CAT_2x: Musculoskeletal system | 0.22 | 0.16 | 0.23 | <0.000001 |
| Discharge Destination: Skilled Care Not Home | 0.06 | 0.11 | 0.05 | <0.000001 |
| Pre-operative serum creatinine | 0.99 | 1.18 | 0.97 | <0.000001 |
| Pre-operative BUN | 16.32 | 18.2 | 16.21 | <0.000001 |
| CPT_CAT_33x-37x: Cardiovascular system | 0.07 | 0.12 | 0.06 | <0.000001 |
| Number of Pulmonary Embolism Occurrences | 0 | 0.03 | 0 | <0.000001 |
| History of severe COPD | 0.04 | 0.09 | 0.04 | <0.000001 |
| Disseminated cancer | 0.02 | 0.06 | 0.02 | <0.000001 |
| Number of Wound Disruption Occurrences | 0 | 0.03 | 0 | <0.000001 |
| Functional health status Prior to Surgery | 0.03 | 0.07 | 0.03 | <0.000001 |
| Pre-operative serum sodium | 138.82 | 138.46 | 138.87 | <0.000001 |
| Steroid use for chronic condition | 0.04 | 0.07 | 0.03 | <0.000001 |
| Currently on dialysis (pre-op) | 0.01 | 0.04 | 0.01 | <0.000001 |
| TRANST_Not transferred (admitted from home) | 0.96 | 0.93 | 0.96 | <0.000001 |
| CPT_Digestive_441x: Intestines—excision | 0.02 | 0.05 | 0.02 | <0.000001 |
| Number of Septic Shock Occurrences | 0.01 | 0.03 | 0 | <0.000001 |
| Wound classification 4: Dirty/Infected | 0.05 | 0.08 | 0.05 | <0.000001 |
| Surgical Specialty: Gynecology | 0.07 | 0.05 | 0.08 | <0.000001 |
| CPT_Cardio_35x: Repairs bypasses etc. | 0.04 | 0.07 | 0.03 | <0.000001 |
| Organ/Space SSI PATOS (Present at the Time of Surgery) | 0 | 0.02 | 0 | <0.000001 |
| CPT_Digestive_48x: Pancreas | 0.01 | 0.03 | 0.01 | <0.000001 |
| CPT_Digestive_49x: Abdomen Peritoneum and Omentum | 0.11 | 0.08 | 0.12 | <0.000001 |
| No dyspnea | 0.05 | 0.08 | 0.05 | <0.000001 |
| Pre-operative International Normalized Ratio (INR) of PT (Prothrombin Time) values | 1.07 | 1.1 | 1.07 | <0.000001 |
| CPT_CAT_60x: Endocrine system | 0.03 | 0.02 | 0.03 | <0.000001 |
| Number of Progressive Renal Insufficiency Occurrences | 0 | 0.02 | 0 | <0.000001 |
| Number of Myocardial Infarction Occurrences | 0 | 0.02 | 0 | <0.000001 |
| CPT_CAT_4x: Digestive system | 0.41 | 0.46 | 0.4 | <0.000001 |
| Number of Unplanned Intubation Occurrences | 0.01 | 0.02 | 0.01 | <0.000001 |
| Discharge Destination: Rehab | 0.03 | 0.05 | 0.03 | <0.000001 |

changed multiple times. To avoid comparisons among variables with a different meaning, we selected only surgeries that took place during 2014. We included only variables that were continuously monitored and used throughout this period; resulting in a total of 231/187 patient variables for post-operative/pre-operative analysis. Patients who died within 30 days of surgery without readmission were excluded. There were a total of 722,101 remaining patients, 39,641 of whom were readmitted within 30 days of discharge, resulting in a readmission rate of 5.49%.

For certain pre-operative lab variables, more than 80% of the entries were missing, and they were excluded from the study. For other variables which had missing data, we used a statistical

method that uses k-nearest neighbors and clustering to find the most likely value for a missing value [16]. The variables were then further separated into two classes: pre-operative variables and post-operative variables. Pre-operative variables are those that can be reliably known before the main surgical procedure starts, whereas post-operative variables (including complications) can only be known after the surgery has occurred. Variable scaling was used for all models, except for Optimal Classification Trees, to bring all values into the range [0,1]; specifically, all variables were normalized by subtracting the minimum and dividing by the range.

## 2.2 Predictive methods

Two different classes of machine learning methods were used. The first class consists of *predictive* methods used to accurately predict the readmission outcome of a patient. Two different scenarios were evaluated: (i) predicting readmissions using pre-operative variables, and (ii) predicting readmissions using both pre-operative and post-operative variables.

We tested a variety of machine learning methods, including: Random Forests (RF) [17], Logistic Regression (LR), Support Vector Machines (SVM), Gradient Boosted Machines (GBM) [18], and Neural Networks (NN) [19]. Logistic Regression aims to fit a regression between the features and the binary outcomes in the logit space. In this study, we fit the logistic regression with a L2-norm regularization term to induce robustness and help guard against data corruption [20]. Both RF and GBM assemble a large collection of classification trees that classifies by taking a majority vote of the individual trees. Linear SVM aims to find a separating hyperplane in the feature space to best separate the patients which were readmitted and those who were not. We implemented a variant of this algorithm, Sparse Linear SVM (SLSVM, see S1 Appendix for details), that chooses a sparse number of variables in the separating hyperplane. By only allowing the hyperplane to depend on a small number of features, we can understand what are the most important variables that separate those that are readmitted and those that are not, which improves interpretability.

To evaluate prediction quality, one typically considers two distinct performance metrics computed out-of-sample: the *false positive rate* (or one minus the *specificity* of the test) and the *true positive rate*, or *sensitivity* of the test. A Receiver Operating Characteristic (ROC) curve evaluates the performance of a binary classifier as the decision threshold is varied, and is formed by plotting the true positive rate against the false positive rate at different threshold settings. To have a single metric to compare different ROC curves, we will consider the Area Under the ROC Curve (AUC). An ideal prediction model has an AUC close to 1, whereas a random prediction would yield an AUC of 0.5.

## 2.3 Prescriptive methods

The second class of methods we employ consists of *prescriptive* methods. We focus on the preoperative hematocrit (HCT) and seek to modulate it in order to minimize the readmission rate. Operationally, this is achieved through blood transfusion before the surgery. Consistent with medical practice, the maximum change in the hematocrit level is limited to 9%, corresponding, roughly, to 3 standard (300cc) bags of blood, which can be considered as a safe upper limit for blood transfusion. Since any such intervention has to be applied before the surgery, the methods we develop will only use pre-operative variables (a total of 88 such variables in the dataset).

We introduce two methods we developed for this study: Prescriptive Support Vector Machines (P-SVM), and Optimal Prescriptive Trees (OPT). They seek to minimize a certain loss function over a set of "actionable" variables that can be controlled through treatments. In this study, the actionable variable is hematocrit, and there are only 4 treatments available: 0%,

3%, 6% and 9% increase in hematocrit, corresponding to 0 to 3 bags of blood transfused. The loss we aim to minimize is the readmission rate.

Before we present our prescriptive methods, we need to establish a baseline for the actionable variable under all treatments. This information would be used by one of our methods (OPT) to learn an effective treatment. In the NSQIP data, we utilize the TRANSFUS variable which indicates whether a pre-operative blood transfusion took place. However, there is no information on the amount of blood transfused. We formed our baseline treatment with the assumption that everyone who has a hematocrit value over 30 had at most 1 bag of blood transfused, as the common operative transfusion threshold is 30 [21]. Then, we add additional bags of blood with decreasing hematocrit levels to bring the patient's hematocrit level above 30. The full table of assumed baseline treatment is shown in Table 1.

The effect of the treatment suggested by our methods will be evaluated using several predictive methods discussed earlier. We rely on four different methods in order to ensure the stability of the result. Specifically, we will use: Random Forests (RF), Logistic Regression (LR), Gradient Boosted Trees (GBM), and Neural Networks (NN). There is evidence in the literature to suggest that pre-operative transfusion could potentially lead to adverse outcomes [22–24]. To ensure such effect is properly accounted for, we additionally consider second order effects of blood transfusion on other pre-operative variables other than hematocrit. Specifically, we fit a regression model of other variables on HCT, and then, calculate how they are affected by the transfusion. We use the modified variables to predict the final readmission outcome.

Prescriptive Support Vector Machines (P-SVM) [25] is an interpretable prescriptive method based on the interpretable SLSVM predictive method we discussed earlier. The method first trains a SLSVM to obtain a hyperplane in a sparse variable subspace that separates readmitted from non-readmitted patients. Fixing this hyperplane, a second optimization problem is formulated, seeking to select the value of the actionable variable (HCT) in order to minimize over the training set a linear combination of the readmission rate and a penalty for changes in the actionable variable. Essentially, this optimization problem sets a value of HCT for each readmitted patient in a way that balances the number of prevented readmissions with the percentage of HCT increase required to prevent them. A detailed mathematical formulation of the method is provided in S1 Appendix.

Optimal Prescriptive Trees (OPT) is an interpretable prescriptive method based on Optimal Classification Trees (OCT). OCTs [26], use integer programming to build a decision tree that optimizes the accuracy of predictions over the training set. A decision tree is interpretable because at each node we are only making a binary decision based on one feature, so the final decision is based upon a series of simple binary decisions. Such a tree, assigns each patient to a leaf node of the tree and makes a prediction for the patient by a majority vote of other patients assigned to the same leaf. OPTs similarly builds an optimal decision tree but with a modified objective, a linear combination of prediction accuracy and the readmission rate. A more detailed mathematical formulation of OPT is in S2 Appendix.

Methods were evaluated in Python, Matlab, and Julia. For random forests, the number of trees grown was 500. Cross-validation was used to tune parameters of the methods.

## 3. Results

### 3.1 Sample characteristics

For each patient, a total of 231 variables were extracted. Table 2 summarizes the baseline demographic and clinical characteristics of the 722,101 patients included in the study. We report the (unnormalized) mean values of the variables over all patients, readmitted patients, and non-readmitted patients, respectively, and only list 60 variables for which the difference

**Table 3. Performance of predictive models.**

| PRE-op Methods | Split I | Split II | Split III | Avg. | Std. |
|---|---|---|---|---|---|
| L2LR | 72.55% | 72.61% | 72.97% | 72.71% | 0.23% |
| SLSVM | 72.51% | 72.58% | 72.91% | 72.67% | 0.21% |
| RF | 73.39% | 73.24% | 73.59% | 73.41% | 0.18% |
| GBM | 73.49% | 73.51% | 73.78% | 73.59% | 0.16% |
| NN | 72.50% | 72.74% | 73.18% | 72.81% | 0.34% |
| **POST-op Methods** | **Split I** | **Split II** | **Split III** | **Avg.** | **Std.** |
| L2LR | 84.20% | 84.36% | 84.64% | 84.40% | 0.22% |
| SLSVM | 84.25% | 84.38% | 84.68% | 84.44% | 0.22% |
| RF | 85.24% | 85.34% | 85.67% | 85.41% | 0.22% |
| GBM | 87.06% | 87.32% | 87.80% | 87.39% | 0.38% |
| NN | 83.03% | 83.06% | 84.00% | 83.36% | 0.55% |

between readmitted and non-readmitted patients was the most statistically significant. Specifically, for each variable we computed a two-tailed p-value using Welch's t-test, where the null hypothesis was that the two cohorts (readmitted and non-readmitted patients) have equal means. Hence, the smaller the p-value, the less likely it becomes that the variable means listed in Table 2 occurred by chance under the null hypothesis. We note that for indicator variables, the means reported correspond to the fraction of patients satisfying the condition.

## 3.2 Accuracy of predictions

For the predictive task, we evaluated the methods across three distinct splits of the data into a training and a test dataset. Each split, randomly selects 80% of the data to form the training set and keeps the remaining 20% as the test set, on which model performance is evaluated. The mean (Avg.) and standard deviation (Std.) of AUC for each predictive method is reported in Table 3; the top table considers predictions using only pre-operative (PRE-op) variables, while the bottom table evaluates models using pre-operative and post-operative variables (POST-op).

Table 3 suggests that readmission predictions are less accurate when using only pre-operative variables. Using all variables, predictions are very strong, achieving an average AUC above 87% (using GBM). Low standard deviations across different splits for all methods, imply that the predictive power is not greatly impacted by the choice of the training data subset. A subgroup analysis using only the general surgery class is contained in S3 Appendix. The results show that models trained on only the subgroup achieved a lower AUC, which suggests that the models trained on the general dataset exhibits favorable cross-learning behavior.

## 3.3 Effectiveness of prescriptions

Predicting readmission is only one step toward preventing readmissions. For the prescriptive task as well, we evaluated the P-SVM and OPT methods across the 3 distinct splits of the data. For each split, we train P-SVM and OPT in the training set and then apply the method to obtain a recommended number of bags of blood to be transfused for each patient whose HCT is less than 30 (HCT<30) in the test set. We evaluate the outcome for each test patient using four different predictive methods: L2LR, RF, GBM, and NN. For each predictive model, we chose a threshold so that the predicted readmission rate equals the ground truth readmission

rate in the training dataset. Such a threshold gives a specificity >96% for all of our models as shown in S4 Appendix. To account for the effects of transfusion on other variables, we modify variables highly correlated with HCT (absolute value>0.1) for each test patient under transfusion using the regression model constructed against HCT. Such variables are: pre-op creatinine, international normalized ratio, prothrombin time, albumin, mortality probability (MORTPROB), and morbidity probability (MORBPROB). We used generalized linear regression models to predict the effect on MORTPROB and MORBPROB since they have bounded values (in [0,1]).

We report in Table 4 the percentage of readmissions prevented in the test set, defined as the ratio (in %) of (i) the number of patients with HCT<30 originally predicted to be readmitted (assuming no treatment) and now predicted not to be readmitted (after treatment), over (ii) the number of patients with HCT<30 predicted to be readmitted (assuming no treatment). We also report the average number of bags of blood per patient under the recommended treatment.

The first column of Table 4 lists the predictive models used to evaluate the effect of treatment, the 2nd and 4rd columns show the percentage of readmissions prevented using the OPT and P-SVM prescriptions, the 3th and 5th columns show the average number of bags per patient when using OPT and P-SVM prescriptions, and the last column reports a baseline percentage of readmissions prevented, assuming any patient (with HCT<30) in the test set gets 1 bag of blood.

We observe that across the different ground truths and splits of the data, the two methods significantly decrease the readmitted patients, on average. For OPT, the average decrease across all splits is 12.15%, while it is 11.45% for P-SVM. Moreover, the average needed bags is about 1 bag, which roughly corresponds to 300cc of blood.

## 4. Discussion

An analysis of the most statistically significant differences in readmitted vs. non-readmitted patients, reveals (cf. Table 2) that the former tend to be patients who underwent vascular surgery, or surgeries involving the pancreas. In contrast, surgeries involving the endocrine system, or Abdomen, Peritoneum, and Omentum are less likely to lead to a readmission. Furthermore, readmitted patients tend to have more complications (e.g., septic shock, bleeding, pneumonia, organ/space/deep incisional SSIs, renal insufficiency) and show higher incidence of return to OR and unplanned intubation.

**Table 4. Percentage reduction of readmissions due to increase in pre-operative hematocrit.**

| Split I | OPT | Averagebags for patients (HCT<30) | PSVM | AverageBags for patients (HCT<30) | decrease_1bag |
|---|---|---|---|---|---|
| LR | 9.27% | 0.97 | 9.49% | 1.06 | 6.78% |
| RF | 14.50% | 0.97 | 14.03% | 1.06 | 9.03% |
| GBM | 4.81% | 0.97 | 3.78% | 1.06 | 3.19% |
| NN | 16.37% | 0.97 | 16.50% | 1.06 | 8.03% |
| Split II | OPT | Averagebags for patients (HCT<30) | PSVM | Averagebags for patients (HCT<30) | decrease_1bag |
| LR | 9.27% | 0.95 | 9.63% | 1.08 | 5.97% |
| RF | 13.08% | 0.95 | 11.30% | 1.08 | 8.24% |
| GBM | 5.34% | 0.95 | 4.20% | 1.08 | 2.84% |
| NN | 18.96% | 0.95 | 19.08% | 1.08 | 9.22% |
| Split III | OPT | Averagebags for patients (HCT<30) | PSVM | Averagebags for patients (HCT<30) | decrease_1bag |
| LR | 10.25% | 0.98 | 10.65% | 1.07 | 6.94% |
| RF | 15.84% | 0.98 | 14.60% | 1.07 | 8.89% |
| GBM | 8.66% | 0.98 | 5.72% | 1.07 | 4.41% |
| NN | 19.51% | 0.98 | 18.37% | 1.07 | 8.46% |

The predictive models we tested lead to very accurate predictions of 30-day readmissions, exceeding 87% AUC with GBM when using all (pre-op and post-op) variables. Using only pre-operative variables, AUC is 74%, on average (with GBM). These results outperform earlier models, such as the LACE index [27], which has an AUC of 68.4%, and more recent models [28], which yield a 72% AUC in 2 days after admission, and 78–81% at discharge.

In terms of specific actionable interventions, we developed prescriptive methods based on the pre-operative predictive models and examined the potential of reducing readmissions by increasing pre-operative HCT levels. We have shown that across a wide variety of different ground truths, two separate prescriptive methods (OPT and P-SVM) are able to prescribe blood transfusion treatments that reduce predicted readmissions for patients with HCT<30, with the decrease ranging from 4.81% to 19.51% for OPT and 3.78% to 19.08% for P-SVM, and with transfusions in the range of 300cc of blood per patient on average. To put the achievable readmission reductions into context, if one could reduce by the mean percentage we achieved (12%) all 30-day readmissions of patients with HCT<30 across the U.S. (over 10,000 per year), the cost savings would amount to $20.3 million on an annual basis [29].

A further potential use of our model is to decrease the length of hospital stay for patients with low-risk of readmission. We can choose a threshold for our models to have high specificity and thus able to accurately identify those that are at low risk of readmission (e.g., the threshold so that the predicted readmission rate equals the ground truth readmission rate in the training dataset as in S4 Appendix).

Moreover, the machine learning methods employed are interpretable. Due to its sparse nature, P-SVM produces a small number of predictive variables which provide an explanation as to why a specific patient has been predicted as having a high readmission risk. For this particular study, P-SVM chose inpatient status, ASA Classification, and mortality probability as some of the variables with most explanatory power on readmission, corresponding to an intuitive understanding that patients that have more critical conditions going into the surgery are more likely to be readmitted. OPTs, additionally, offer the ability to examine every prediction or prescription and identify the specific path through the decision tree that led to the decision. Fig 1 depicts a part of a prescriptive tree. At every branching step, a binary decision is made.

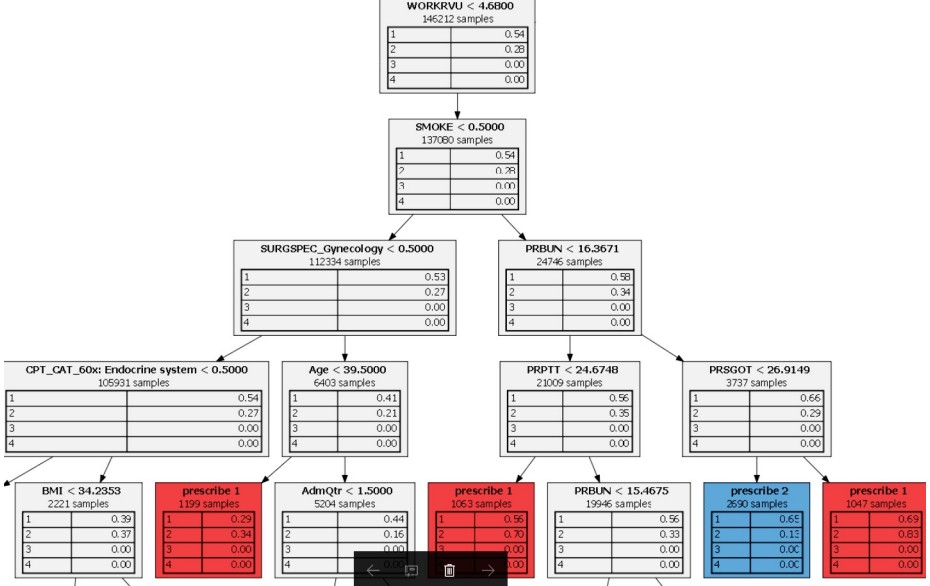

**Fig 1. An instance of an Optimal Prescriptive Tree.**

For example, at the top node, the patients are split into two classes, smoking or non-smoking. Several splits lead to the leaves of the tree (colored red and blue in this case) in which all patients corresponding to a leaf are prescribed a certain treatment. Prescription 1 corresponds to no transfusion and Prescription 2 corresponds to transfusing one bag of blood. Interpretable decision trees enable doctors and experts to understand the proposed decisions and, potentially, further improve the model based on the binary decisions it makes.

A limitation of our study is the lack of the full ground truth for the impact of the proposed interventions, namely, whether they prevent readmissions. That is impossible to ascertain without performing a randomized clinical trial. Instead, we use a number of strong predictive models to evaluate the impact of the derived prescriptions and observe consistent readmission rate reduction across these models. We hope that this work motivates clinical trials that could confirm our findings. Furthermore, beyond the short-term readmission outcome, it would be beneficial to understand how these measures can impact the long-term oncologic outcome, which is equally if not more important. However, the NSQIP database unfortunately does not track patients long term, and thus we do not have the required data to conduct such an analysis in the paper.

## 5. Conclusions

We leveraged a large national dataset of surgical patients with the goal of reducing 30-day readmissions. We developed both predictive and prescriptive machine learning models. The former predict 30-day readmissions and identify the most discriminative variables. The latter, build on the predictive models and can offer specific recommendations on actionable decisions to reduce readmissions. We focused on pre-operative hematocrit and showed how to make personalized recommendations to increase its value, when needed, through blood transfusion.

Prediction accuracy with our methods exceeds 87% using the entire collection of NSQIP variables and 74% using only variables known pre-operatively. The proposed prescriptions/interventions can reduce the predicted readmissions by 11.45%-12.15% for patients with HCT<30, on average. Beyond improving patient outcomes, this reduction can lead to more than $20 million in annual savings in the U.S.

## Supporting information

**S1 Appendix.**
(DOCX)

**S2 Appendix.**
(DOCX)

**S3 Appendix.**
(DOCX)

**S4 Appendix.**
(DOCX)

## Acknowledgments

The authors would like to thank Dr. George Velmahos at the Massachusetts General Hospital and Dr. George Kasotakis at the Boston Medical Center for useful discussions and for providing access to the NSQIP data.

## Author Contributions

**Conceptualization:** Ioannis Ch. Paschalidis.

**Formal analysis:** Michael Lingzhi Li, Taiyao Wang.

**Funding acquisition:** Ioannis Ch. Paschalidis.

**Investigation:** Michael Lingzhi Li, Ioannis Ch. Paschalidis, Taiyao Wang.

**Methodology:** Dimitris Bertsimas, Michael Lingzhi Li, Ioannis Ch. Paschalidis, Taiyao Wang.

**Project administration:** Ioannis Ch. Paschalidis.

**Resources:** Ioannis Ch. Paschalidis.

**Software:** Michael Lingzhi Li, Taiyao Wang.

**Supervision:** Dimitris Bertsimas, Ioannis Ch. Paschalidis.

**Validation:** Michael Lingzhi Li, Taiyao Wang.

**Visualization:** Michael Lingzhi Li, Taiyao Wang.

**Writing – original draft:** Michael Lingzhi Li, Ioannis Ch. Paschalidis, Taiyao Wang.

**Writing – review & editing:** Dimitris Bertsimas, Ioannis Ch. Paschalidis.

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
