## [Decision Letter · Decision Letter 0]

8 Jun 2020

PONE-D-20-06414

Prescriptive Analytics for Reducing 30-day Hospital Readmissions after General Surgery

PLOS ONE

Dear Dr. Paschalidis,

Thank you for submitting your manuscript to PLOS ONE. After careful consideration, we feel that it has merit but does not fully meet PLOS ONE’s publication criteria as it currently stands. Therefore, we invite you to submit a revised version of the manuscript that addresses the points raised during the review process.

This study is a good try in this field. We look for your revision in response to some clinically relevant comments by reviewers and editors. 

We look forward to receiving your revised manuscript.

Kind regards,

Chun Chieh Yeh, M.D., Ph.D.

Academic Editor

PLOS ONE

Journal Requirements:

Additional Editor Comments (if provided):

The manuscript is a good try to apply machine learning to develop a predicting model to reduce 30-day readmission after general surgery. However, some clinically relevant issues need to be responded appropriately in revised edition.

In addition to the reviewer's comments, I have some two concerns about this study.

1. Blood transfusion is well known in association with post-operative recurrent cancer in oncologic surgeries. I think long-term oncologic outcome should be more important than shot-term readmission rate. Could you use your dataset to study the influence of pre-operative blood transfusion in post-operative oncologic outcomes (e.g. disease free survival ) after oncologic surgeries (e.g. pancreas surgeries for pancreatic cancer or liver resection for liver cancer? or colon surgeries for colon cancer?)? OR, at least, you should address this in your discussion part.

2. Please revise your abstract to fit the format for publication (i.e., Introduction /Material and method/ Result/ Conclusion )

Reviewers' comments:

Reviewer's Responses to Questions

**Comments to the Author**

1. Is the manuscript technically sound, and do the data support the conclusions?

Reviewer #1: Yes

Reviewer #2: Yes

2. Has the statistical analysis been performed appropriately and rigorously? 

Reviewer #1: Yes

Reviewer #2: Yes

3. Have the authors made all data underlying the findings in their manuscript fully available?

Reviewer #1: Yes

Reviewer #2: Yes

4. Is the manuscript presented in an intelligible fashion and written in standard English?

Reviewer #1: Yes

Reviewer #2: Yes

5. Review Comments to the Author

Reviewer #1: This study uses NSQIP data and machine learning methods to predict 30-day readmissions and recommend pre-operative blood transfusions to reduce readmissions by 12%. We are happy to see using AI in this field.

1.Do you consider to separate different surgery type in analysis? or do subgroup analysis for different surgery type? (for example: abdominal surgery/non-abdominal surgery). I think it may increase AUC according to previous experience

2.Readmission is closely correlative with many factors in the previous hospitalization, especially post-operative outcome. Did you try to do predictive model using only post-operative variables?

3. In the aspect of clinical use, another possible application of this study is to decrease the length of hospital stay (previous hospitalization) in low risk patient. Do you evaluate the specificity(or false negative rate) of this model?

4.How do you think about 87% and 74% AUC in AI study? Do you estimate if we want to achieve enough benefit(reduce readmission and save money), how higher AUC do we need?

5. You did many machine learning methods in the study and also mention about “interpretable”. Please try to add more discussion about the machine learning methods and you result. It needs more integration (and explanation) of different machine learning methods, results and your clinical problems.

Reviewer #2: THE DATA SHOWED a large national dataset of surgical patients with the goal of reducing 30-day readmissions. hOW TO INTERPRETED THE RELATIONSHIP BETWEEN BLOOD TRANSFUSION AND READMISSION , FOCUSSING ON THE LISTED COMPLICATION, SUCH AS SEPTIC SHOCK , BLEEDING PNEUMONIA , SSIs, RENAL INSUFFICIENCY

6. PLOS authors have the option to publish the peer review history of their article (what does this mean?). If published, this will include your full peer review and any attached files.

Reviewer #1: Yes: Chien Hui Wu, MD

Reviewer #2: Yes: Ming-Chin Yu

---

## [Author Response · Author response to Decision Letter 0]

17 Jul 2020

Please see attached Response Letter.

---

## [Decision Letter · Decision Letter 1]

11 Aug 2020

Prescriptive Analytics for Reducing 30-day Hospital Readmissions after General Surgery

PONE-D-20-06414R1

Dear Dr. Paschalidis,

We’re pleased to inform you that your manuscript has been judged scientifically suitable for publication and will be formally accepted for publication once it meets all outstanding technical requirements.

Kind regards,

Chun Chieh Yeh, M.D., Ph.D.

Academic Editor

PLOS ONE

Additional Editor Comments (optional):

Your work is good for acceptance in its current content after scientific reviewing.

Reviewers' comments:

Reviewer's Responses to Questions

**Comments to the Author**

1. If the authors have adequately addressed your comments raised in a previous round of review and you feel that this manuscript is now acceptable for publication, you may indicate that here to bypass the “Comments to the Author” section, enter your conflict of interest statement in the “Confidential to Editor” section, and submit your "Accept" recommendation.

Reviewer #1: All comments have been addressed

2. Is the manuscript technically sound, and do the data support the conclusions?

Reviewer #1: Yes

3. Has the statistical analysis been performed appropriately and rigorously? 

Reviewer #1: Yes

4. Have the authors made all data underlying the findings in their manuscript fully available?

Reviewer #1: Yes

5. Is the manuscript presented in an intelligible fashion and written in standard English?

Reviewer #1: Yes

6. Review Comments to the Author

Reviewer #1: Good revision. The article is more close to the clinical use now. Hope to see more AI study using NASQIP data

7. PLOS authors have the option to publish the peer review history of their article (what does this mean?). If published, this will include your full peer review and any attached files.

Reviewer #1: **Yes: **Chien-hui Wu